# Thermomechanically Induced Precipitation in High-Performance Ferritic (HiperFer) Stainless Steels

**Xiuru Fan [1], Bernd Kuhn [1,\*] , Jana Pöpperlová [2] , Wolfgang Bleck [2] and Ulrich Krupp [2]**

1   Institute of Energy and Climate Research (IEK), Microstructure and Properties of Materials (IEK-2), Forschungszentrum Juelich GmbH, 52425 Jülich, Germany; x.fan@fz-juelich.de
2   Steel Institute RWTH Aachen University (IEHK), 52072 Aachen, Germany; Jana.Poepperlova@iehk.rwth-aachen.de (J.P.); bleck@iehk.rwth-aachen.de (W.B.); Krupp@iehk.rwth-aachen.de (U.K.)
\*   Correspondence: b.kuhn@fz-juelich.de; Tel.: +49-2461-61-4132

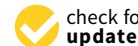

**Featured Application: Forged high-temperature components such as bolts, nuts and blades for turbines, compressors, and pumps with increased resistance to fatigue, creep, and corrosion; eventually, tools applied in techniques for thermoforming of metals like stamping and deep drawing.**

**Abstract:** Novel high-performance fully ferritic (HiperFer) stainless steels were developed to meet the demands of next-generation thermal power-conversion equipment and to feature a uniquely balanced combination of resistance to fatigue, creep, and corrosion. Typical conventional multistep processing and heat treatment were applied to achieve optimized mechanical properties for this alloy. This paper outlines the feasibility of thermomechanical processing for goal-oriented alteration of the mechanical properties of this new type of steel, applying an economically more efficient approach. The impact of treatment parameter variation on alloy microstructure and the resulting mechanical properties were investigated in detail. Furthermore, initial optimization steps were undertaken.

**Keywords:** HiperFer steel; thermomechanical processing; Laves phase; precipitation

## 1. Introduction

Fully ferritic, Laves-phase strengthened [1] stainless steels with high chromium content show potentially sufficient resistance to steam oxidation at operating temperatures as high as 650 °C [2–5]. Based on this knowledge, advanced High-performance Ferritic (HiperFer) stainless steels were designed for application in the next generation of supercritical thermal power plants [5–10]. The heightened operation parameters (650 °C/300 bar) of these plants necessitate increased resistance to creep of the applied structural materials up to 650 °C, in combination with sufficient steam-oxidation resistance [3,11]. Furthermore, demands concerning the fatigue resistance of structural power-plant materials are rising due to the increasing implementation of regenerative, intermittent sources of power in the grid. Current state-of-the-art advanced ferritic–martensitic (AFM) steels, with 9–12 wt. % chromium content, are limited to application temperatures below 620 °C because of either limitations in steam-oxidation resistance (9 wt. % Cr) or a drop in long-term creep strength (12 wt. % Cr) [12–17]. Novel HiperFer steels potentially fulfill these mechanical strength requirements via a combination of solid-solution strengthening and the precipitation of finely dispersed intermetallic $(Fe,Cr,Si)_2(Nb,W)$ Laves-phase particles [4–9,18]. Furthermore, steels with high chromium content provide superior steam-oxidation resistance [2,3].

Several variants of HiperFer steel were described by Kuhn et al. [4–10]. As displayed in Figure 1, the phase fraction of the strengthening Laves phase increased with higher alloying contents of W and

Nb. While the low-alloyed HiperFer base variant 17Cr2 (17Cr2.6W0.6Nb) needs precipitation annealing (PA) to exploit its maximum strength potential [6,19], the 17Cr5 steel (17Cr4W1Nb) constitutes an advanced, age-hardening variant, which was developed to be put into service in its recrystallized state without a significant loss in creep or fatigue strength [8].

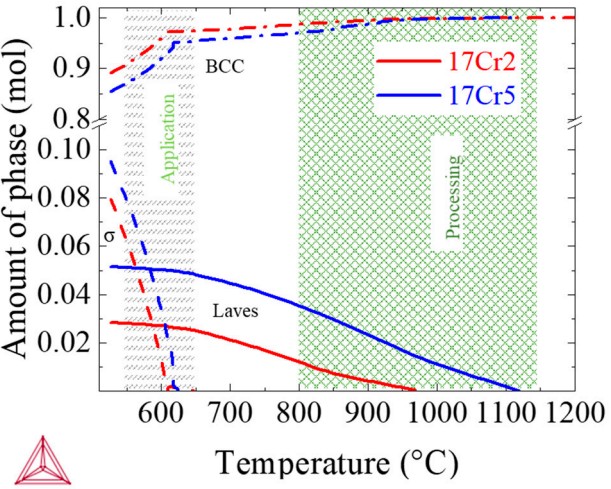

**Figure 1.** Area fraction of Laves phase in two HiperFer steels (17Cr2:17Cr2.6W0.6Nb0.2Si, 17Cr5:17Cr4W1Nb0.2Si).

Typically, the HiperFer base variant (17Cr2) is recrystallized after forming by forging or rolling, followed by multistep processing (MSP), i.e., precipitation annealing (PA) [8,20]. This multistep annealing must not be confused with the standard-quality heat treatment of ferritic–martensitic 9–12Cr steels, where stress from martensitic transformation is reduced first and precipitation annealing is done in a (or multiple) consecutive step(s), typically at temperatures exceeding the envisaged application temperature range [13,21,22]. HiperFer is designed to be precipitation-annealed for creep strength at temperatures lower than (cf. "Temp L" in Figure 2a) and/or at (cf. "Temp H" in Figure 2a) application temperature, thus potentially enabling in situ PA during commissioning of a power plant or component [8]. Nevertheless, such multistep processes are time-consuming and thus cost-intensive if performed ex situ (i.e., prior to application). Deformation has a beneficial impact on the precipitation of Laves-phase particles in these types of steel [23] and leads to "reactive strengthening" [8,24,25]. Integrated thermomechanical processing (TMP; cf. Figure 2b), incorporating forming and PA in the same process, would therefore greatly improve cost efficiency. In such an integrated process (details given in Reference [26]) the material is homogenized after casting and thickness is reduced by forging and/or rolling outside the precipitation temperature range. Subsequently, the material is recrystallized, reduced to final thickness by controlled forging/rolling (and optional short holding) inside the precipitation temperature range, followed by rapid cooling. This integrated processing route induces precipitation of the strengthening Laves-phase particles during final thickness reduction (and optional holding), effectively reduces processing time and heating energy, and thus improves throughput in production at lowered cost. The 17Cr5 alloy provides a suitable temperature window for integrated processing (governed by the dissolution temperature of the Laves phase; cf. Figure 1). With the use of integrated processing, this high-alloyed variant may reach mechanical properties that at least rival the conventional multistep-processed 17Cr2 base alloy.

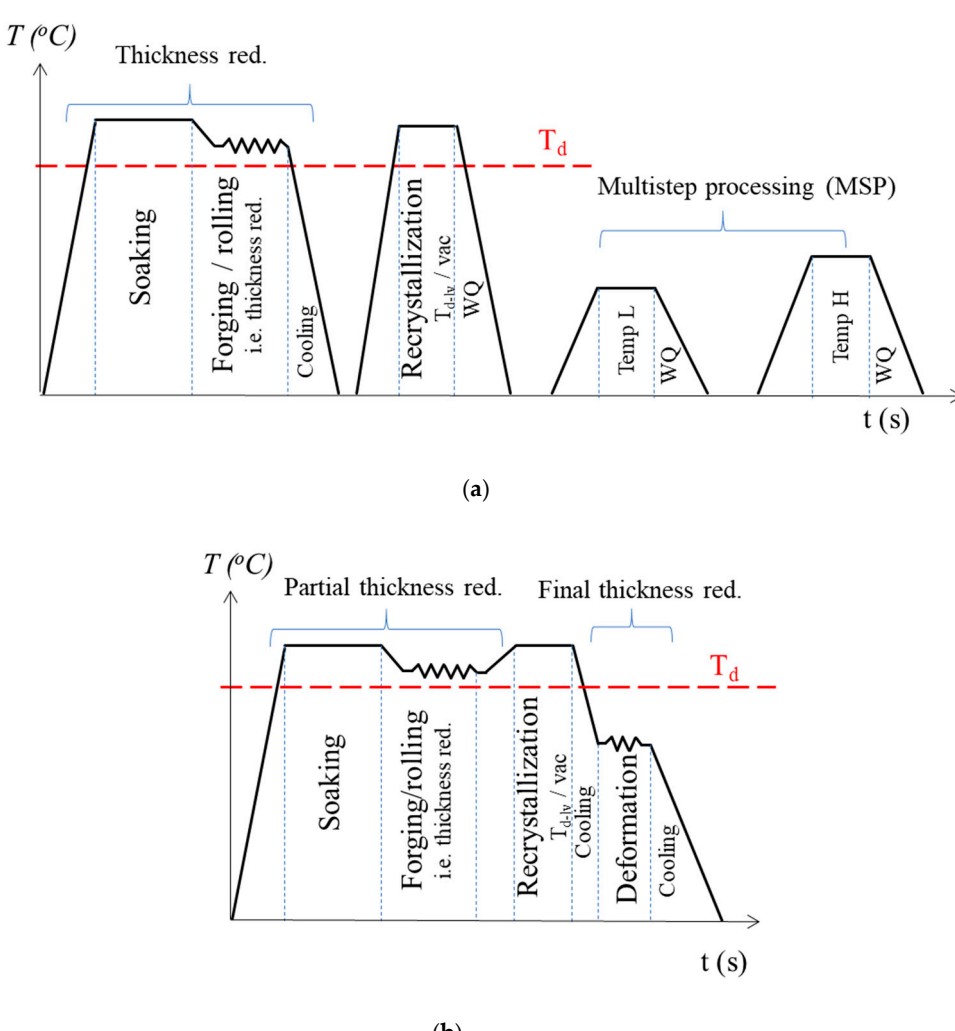

**Figure 2.** Processing of HiperFer steel via (**a**) standard multistep processing (MSP) and (**b**) integrated thermomechanical processing (TMP). $T_d$: Dissolution temperature of the Laves phase.

## 2. Materials and Methods

### 2.1. Alloy Production and Processing

Two material batches, named 17Cr5_1 and 17Cr5_2 (actual chemical compositions stated in Table 1), were vacuum-induction-melted from high-purity raw materials, soaked at 1250 °C for 2 h, and air-cooled. Specimens of $15 \times 15 \times 65$ mm$^3$ in size were cut from the homogenized blocks to be used for simulated TMP.

**Table 1.** Chemical compositions of the trial alloys (wt. %)

| Batch ID: | Cr | W | Nb | Si | Mn | Fe | C | N |
|---|---|---|---|---|---|---|---|---|
| **17Cr5_1** | 17.43 | 3.85 | 0.9 | 0.25 | 0.25 | 77.33 | <0.01 | <0.001 |
| **17Cr5_2** | 17.30 | 3.97 | 0.93 | 0.27 | 0.23 | 77.40 | <0.01 | <0.001 |

To simulate integrated thermomechanical processing at the laboratory scale, a Bähr TTS 820 thermomechanical treatment simulator was utilized at the Steel Institute (IEHK) of the Northrhine-Westfalian Technical University (RWTH), Aachen. Simulated TMP incorporated recrystallization/dissolution annealing at 1200 °C for 40 min [8], followed by a single-step deformation simulating the final deformation step of the integrated thermomechanical treatment

route. The preliminary thickness-reduction steps involved in real-life processing were not covered by this study, because these would have been extinguished by recrystallization (i.e., prior to the final thickness-reduction step, cf. Figure 2b). Table 2 summarizes the variations of deformation temperature ($T_\varphi$), deformation degree ($\varphi$), and holding time after deformation ($t_H$). The TMP parameters were intended to create a rise in dislocation density to, in turn, induce boosted particle strengthening via increased precipitation of Laves particles of suitable size and spacing. Laves-phase particles pin the dislocations they precipitate on, and thus make nonpermanent dislocation strengthening (at least partly) permanent [27].

**Table 2.** TMP parameters.

| $T_\varphi$ (°C) | $\varphi$ (−) | $t_H$ (s) |
|---|---|---|
| **800, 950** | 0, 0.2, 0.5 | 0, 60, 80, 300 |

### 2.2. Microstructure Observation

Samples were cut from the deformed parts of the TMP specimens and cold-mounted in epoxy resin for sample preparation. The mounted samples were ground and polished (diamond polishing solution down to 1 μm surface roughness, followed by a $SiO_2$ solution used for vibration polishing as the final step). After preparation, samples were etched in ethanol/$H_2SO_4$ solution for 3 to 5 s. Images with a resolution of $6144 \times 4608$ pixels were taken using a Zeiss Merlin high-resolution field emission scanning electron microscope (HR-FESEM). Applying the image analysis method reported by Lopez et al. [28–31], these images were analyzed quantitatively, utilizing the commercial software package analySIS pro. A Zeiss Auriga focused ion beam (FIB) machine was applied for the preparation of lamellae for transmission electron microscopy (TEM; Zeiss Libra 200 TEM, operating voltage 200 kV).

### 2.3. Mechanical Testing

In order to quantify the impact of the changes implemented in MSP and TMP, ambient-temperature mechanical testing was performed. Because of the differences in dimensions and thermal/thermomechanical treatment methods of the studied materials, tensile experiments were performed for standard uniaxial full-size and flat miniature-size specimens. Standard full-size uniaxial tensile specimens (gauge dimensions: 30 mm in length, 6.4 mm in diameter) were cut via electric-discharge machining and subsequent turning from MSP-treated material. In the case of the TMP materials, flat miniature tensile specimens ($10 \times 2 \times 1$ mm$^3$ in gauge dimensions) were cut via electrical-discharge machining after trial processing. The deformed area of the TMP specimen was located within the gauge length of the resulting flat miniature tensile specimen, while the specimen shoulders were located in the nondeformed grip parts of the initial TMP specimen. Ambient-temperature tensile testing was carried out by applying a strain rate of $10^{-3}$/s, utilizing Instron 1362 and Instron 8862 testing machines.

## 3. Results and Discussion

### 3.1. Mechanical Testing

Figure 3 presents the tensile properties of the different TM-processed 17Cr5 model alloys in comparison to the 17Cr2 base alloy in the MSP state [8,9]. Because of a lack of strengthening precipitates, the ultimate tensile strength (UTS) of the recrystallized (RX: 1100–1125 °C/25 min [8]) 17Cr5 alloy was about 470 MPa, with an offset yield strength ($Ys_{0.2}$) of approximately 370 MPa. After standard MSP treatments (MSP1: RX + 530 °C/2−5 h/WQ + 600−650 °C/0.5−5 h/WQ; MSP2: RX + 540 °C/5−10 h/WQ + 600−650 °C/5−10 h/WQ), the tensile strength was almost doubled with sufficient rupture elongation (Figure 3).

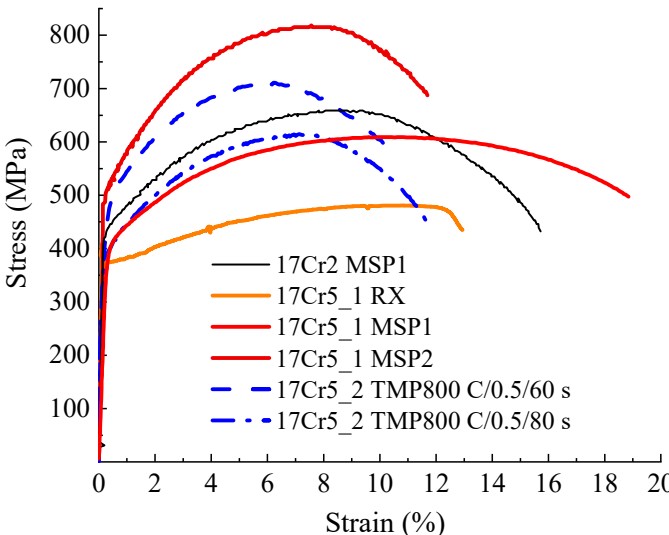

**Figure 3.** Ambient-temperature stress–strain curves of MSP and TMP 17Cr5 alloys in comparison to the base 17Cr2 alloy in the MSP state.

The 17Cr5_2 TMP material (parameters: T: 800 °C, φ: 0.5, $t_H$: 60 s, cf. Figure 3) reached a UTS of approx. 710 MPa with a 0.2% offset yield strength ($YS_{0.2}$) of 495 MPa. Despite the limitations of miniature specimen geometry and comparatively large grain size (800 to 1500 μm), a promising rupture elongation of about 10% was achieved. Prolonged holding (80 s) after deformation did not yield improved UTS, but did yield an increase in rupture elongation. Considering specimen geometry, i.e., gauge cross-section (2 mm × 1 mm), in relation to grain size, the measured rupture elongations were quite conservative. For uniaxial testing specimens with adjusted grain size (200–300 μm), higher elongation values can be expected [32]. Nevertheless, economic, integrated thermomechanical processing can provide tensile properties close to those achieved via standard multistep processing. The microstructures of the TMP and MSP materials were observed via HR-FESEM (cf. Figure 4). Typical microstructures from MSP 17Cr5 steel (schematic in Figure 4a) are displayed in Figure 4b,c. Microstructures from TMP steel (800 °C/0.5/60 s) are depicted in Figure 4d,e.

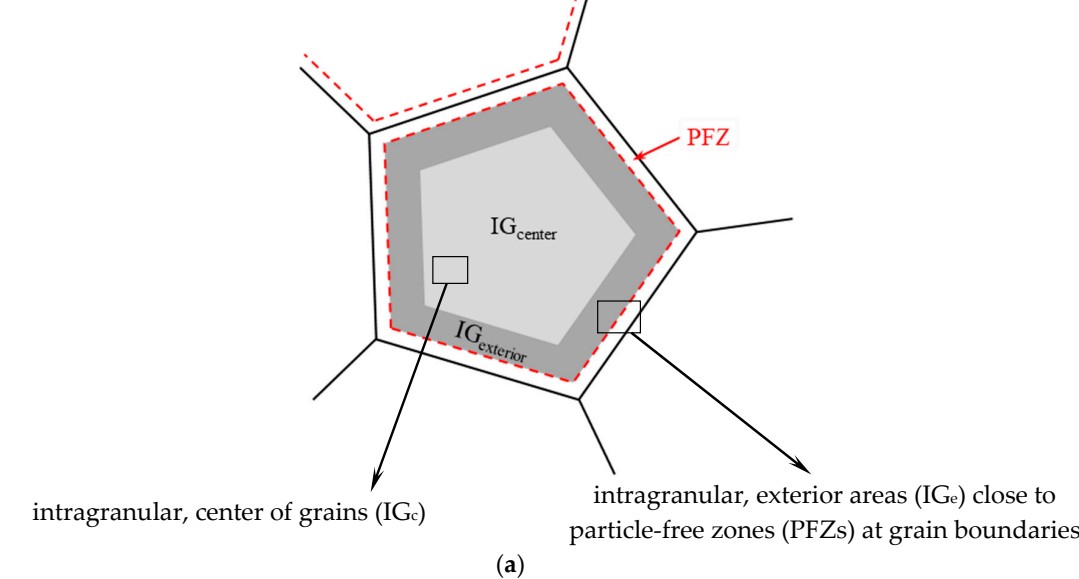

**Figure 4.** *Cont.*

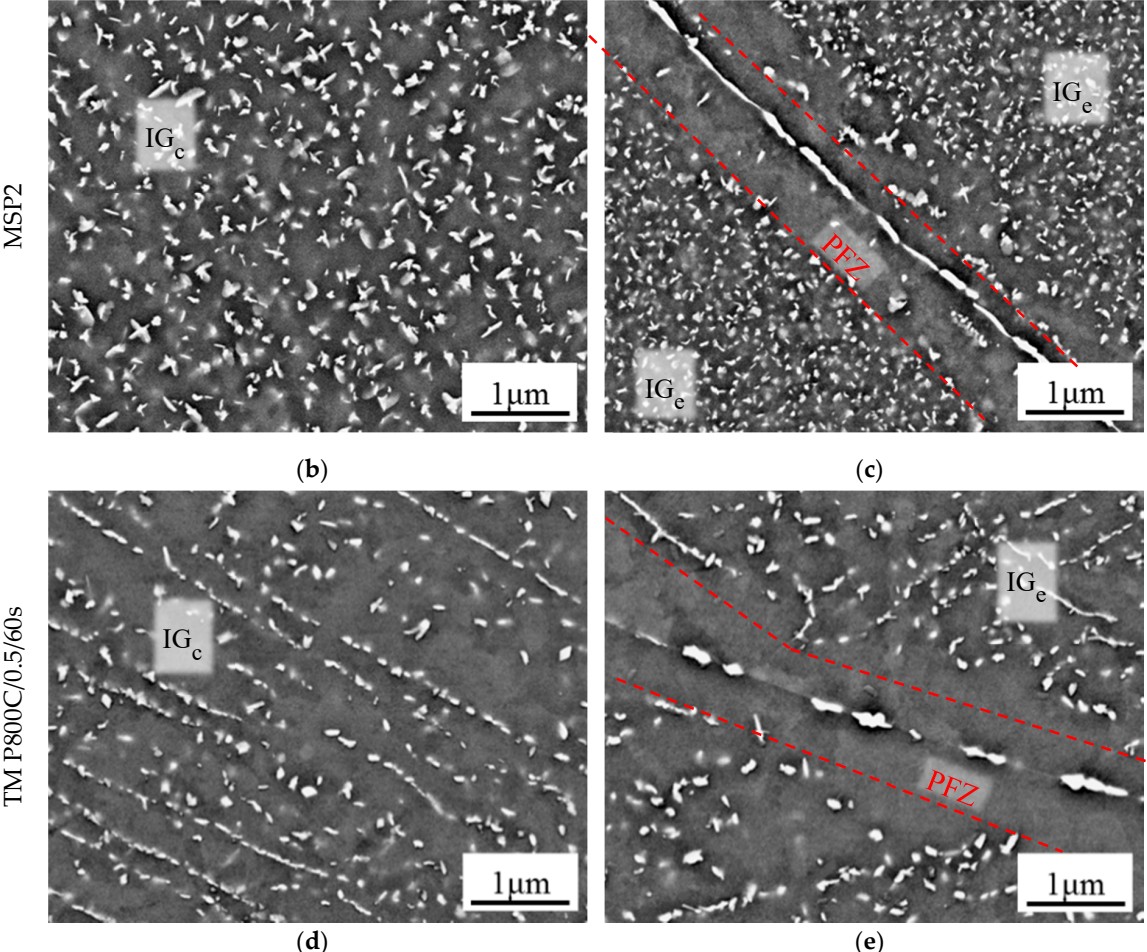

**Figure 4.** Schematic drawing of precipitate microstructure (**a**). SEM micrographs of 17Cr5 steel: Precipitate morphology in the grain interiors (IG$_c$) after MSP2 (**b**) and TMP800C/0.5/60s (**d**) and intragranular particle morphology in exterior areas (IG$_e$) close to PFZs in MSP2 (**c**) and TMP800C/0.5/60s (**e**) materials.

Precipitation of a multitude of small Laves-phase particles was obviously induced by both MSP and TMP. The particle coverage of high-angle grain boundaries appeared more complete in MSP specimens, because of pronounced diffusion-controlled precipitate growth [10,33,34]. The enrichment of key elements for Laves-phase nucleation at grain boundaries decreased the concentration of these in the surrounding matrix and led to delayed nucleation and growth of the Laves-phase particles. In turn, this caused the formation of characteristic particle-free zones (PFZs) along grain boundaries [28–30] and a tendency towards larger particles towards the grain interiors. As a consequence, the intragranular precipitates in the MSP material had a bimodal size distribution and were typically larger in the centers (cf. Figure 4b; Figure 5a, MSP IG$_c$) than in the exterior (near PFZs) areas of grains (cf. Figure 4c; Figure 5a, MSP IG$_e$). Intragranular particles, which are located at the exteriors of grains close to the PFZs (IG$_e$; cf. Figure 4c, Figure 5a: MSP IG$_e$) and the PFZ width itself were smaller (appr. 35–40 nm) in the case of the MSP material. With a plateau from 20 to 100 nm in the size distribution, the TMP material, in contrast, did not display such pronounced interrelations of grain boundaries, PFZs, and particle size (cf. Figure 4d,e; Figure 5a, TMP800C/0.5/60s IG$_c$ vs. TMP800C/0.5/60s IG$_e$).

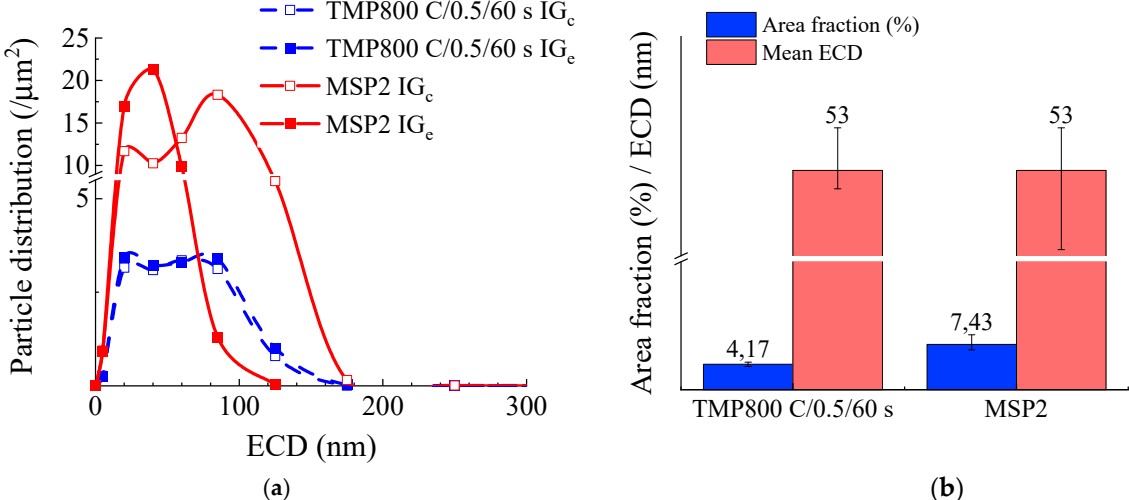

**Figure 5.** Quantitative image analysis results of (**a**) particle size distribution and (**b**) averaged particle area fractions against averaged ECDs (equivalent circle diameter) of intragranular precipitates.

In terms of deformation, TMP induces additional dislocations and thus may alter the energy levels of the grain interiors and the grain boundaries near areas, which in turn may result in a more homogenous particle size distribution. Furthermore, the comparably high temperature of TMP accelerates the growth of precipitates. Nevertheless, TMP yielded an average particle size comparable to that of MSP (Figure 5b, ECD). While the intragranular particles were homogenously distributed in the MSP material, TMP produced a preferentially linear alignment of Laves-phase particles (Figure 4, bottom).

The higher particle density of the MSP material (Figure 5a) provided a higher tensile strength. Nevertheless, even with just one fifth of the particle density (Figure 5a) and about half the particle area fraction (Figure 5b), the TMP-treated alloy, most probably because of the increased dislocation density from TMP deformation in comparison to the MSP material, yielded a UTS that was just 10% lower at ambient temperature (Figure 3).

Because of the comparable tensile strength values of the material TMP-treated at T = 800 °C, $\varphi$ = 0.5, and prolonged holding time of $t_H$ = 80 s (cf. Figure 3) and the MSP1 material (cf. Figure 3), the microstructures of these two variants were compared in detail by TEM analysis (Figure 6). FIB lamellae were cut perpendicular to the linearly aligned precipitates (cf. "plane of view" indicated by arrows in the left column of Figure 6) in the case of the TMP variant. At the same tensile and offset yield strength values, the TMP800C/0.5/80s material presented much larger precipitates than the MSP1 material did. Shear bands with clusters of dislocations and particles precipitated along them were observed in the TMP800C/0.5/80s lamellae. A higher dislocation density at shear bands favors the formation of Laves-phase particles and led to the observed linear alignment of precipitates. Both treatments mainly provided largely disc-shaped particles (Figure 6, left column). Volumetric expansion into all three dimensions was observed, with sizes ranging from 40 to 400 nm in diameter and 20 to about 80 nm in thickness in the TMP800C/0.5/80s variant. In contrast to this, both needle-shaped and comparably thin, disc-shaped particles were observed in the MSP1 lamellae, which may have been caused by observation of thin disc-shaped particles from different relative orientations. In former studies, Laves-phase particles mainly presented volumetric expansion into two dimensions, leading to disc shapes of typically less than 20 nm in thickness and 20 to 200 nm in diameter [4,8,18,28–31]. The TMP800C/0.5/80s material provided tensile properties in the range of the MSP1 material, which obviously presented more, finer, and more homogenously distributed particles. As expected, the thermomechanically processed TMP800C/0.5/80s material not only provided dispersion, but also combined dislocation strengthening, which obviously compensated the drawback in dispersion hardening.

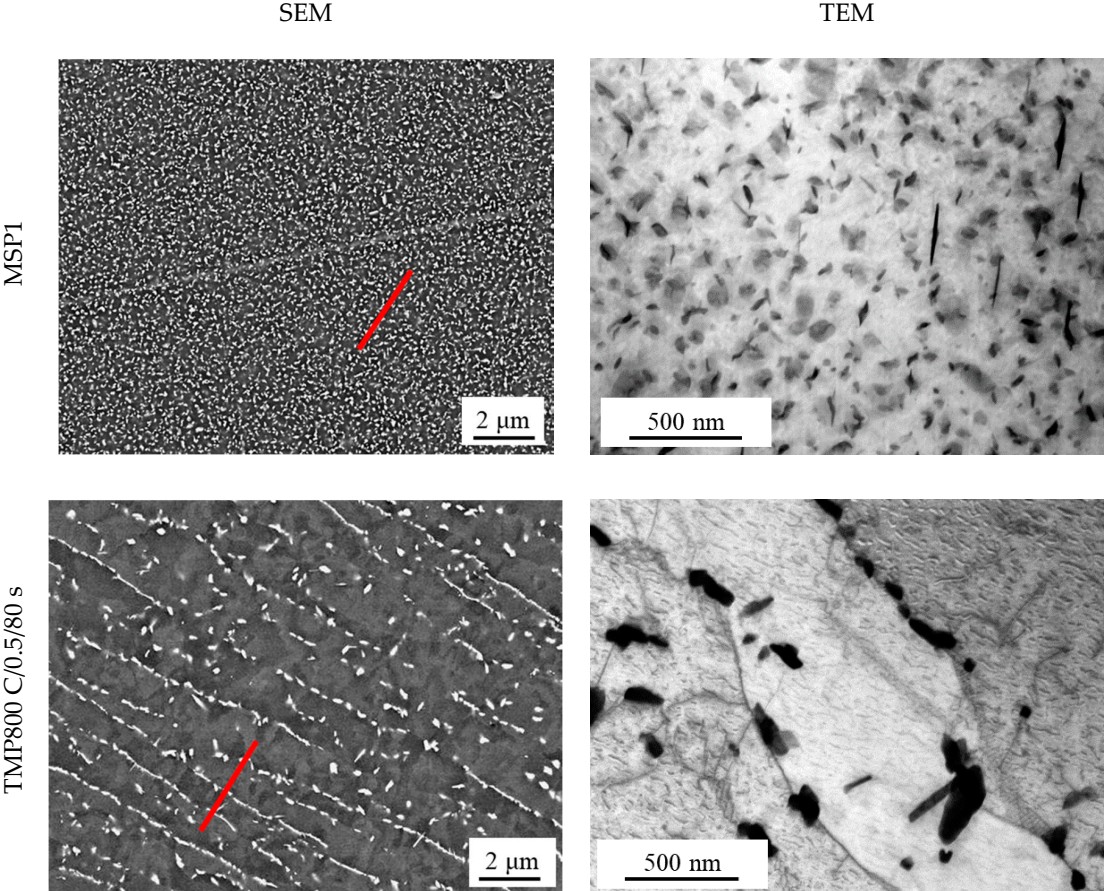

**Figure 6.** SEM (**left**, arrows indicate plane of TEM view)/TEM (**right**) micrographs: intragranular precipitate morphology (IG$_c$) of MSP1 (**top**) and TMP800C/0.5/80s (**bottom**) materials.

*3.2. Impact of Deformation Parameter Variation*

3.2.1. Temperature

As demonstrated, the microstructure and mechanical properties of HiperFer alloys are readily adjustable using thermal and/or thermomechanical processing. For this reason, the impact of systematic variations in processing parameters (cf. Table 2) on the mechanical properties of the materials was studied. The temperature of deformation was varied from 800 °C to 950 °C. Figure 7a displays the corresponding changes in particle density and area fraction, derived from image analysis, and illustrative inlay micrographs. At 800 °C, the particles were clearly arranged in arrays, which was less apparent at 950 °C. While the area fraction of the Laves-phase precipitates remained stable (within the range of scatter), the particle density decreased significantly with increasing deformation temperature (deformation degree $\varphi$ = 0.5, holding time $t_H$ = 60 s). This was caused by a pronounced particle coarsening at the higher processing temperature (950 °C). Figure 7b displays the variation in size distribution of particles with increasing deformation temperature. At 800 °C, the largest ECD measured was less than 180 nm, with a plateau from values around 20 up to 80 nm. With increasing temperature during TMP0.5/60s processing, the maximum ECD value increased to about 400 nm. Most of the particles found had ECD sizes of around 125 nm and the plateau in distribution disappeared. A higher processing temperature implies higher diffusion velocities of Nb and W [35,36], the main Laves-phase-forming elements, which boosts precipitation and subsequent particle growth kinetics. Despite the stable area fraction, the increase in particle diameter caused by higher deformation temperature led to a reduction in tensile strength but an increase in ductility (Figure 7c). This demonstrates how important it is

to not draw conclusions based exclusively on particle area fraction results. For decisive evaluation, a comprehensive analysis of both area fraction and size distribution is required.

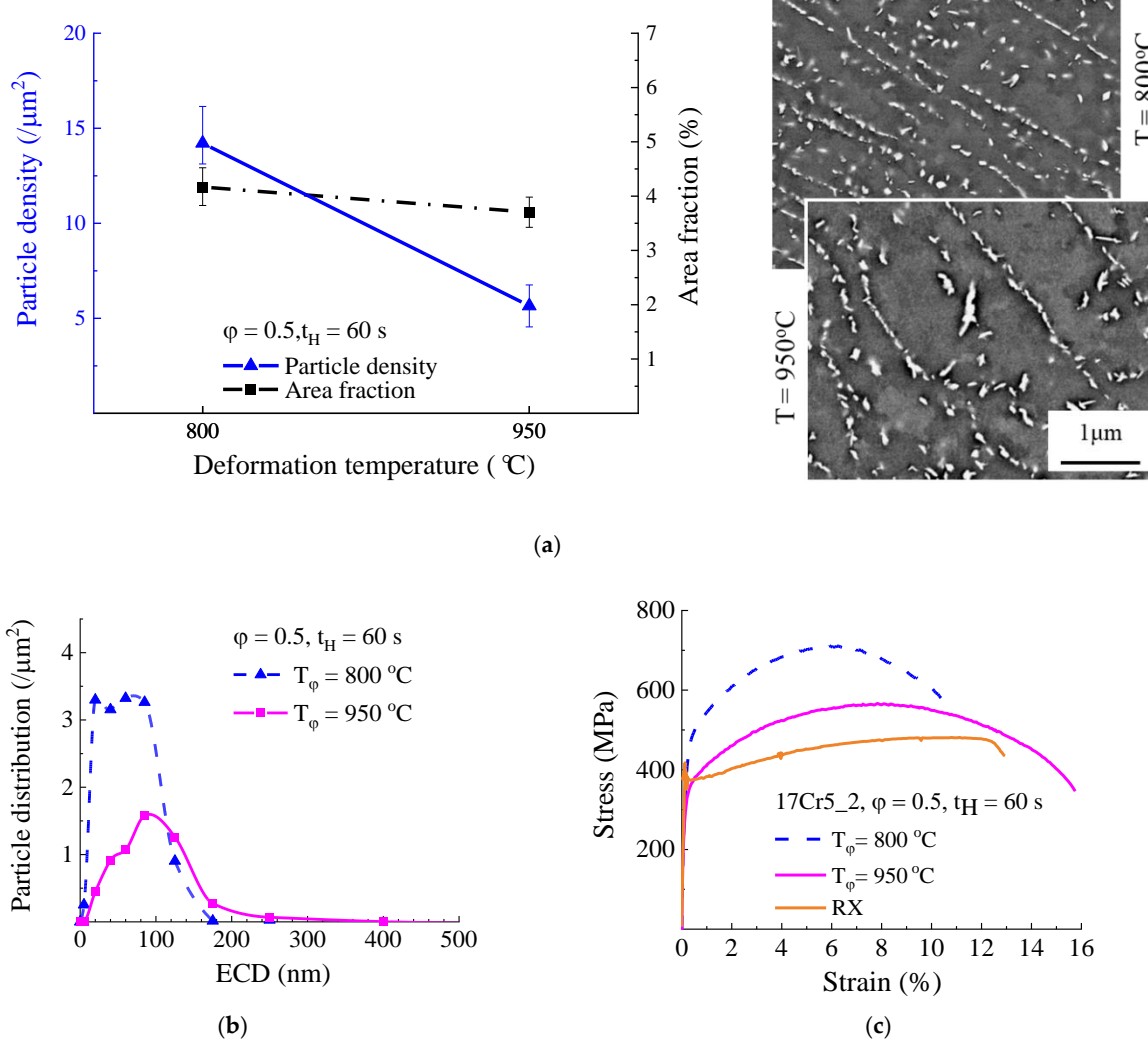

**Figure 7.** Impact of TMP temperature on (**a**) particle density and area fraction, (**b**) particle size distribution, and (**c**) ambient-temperature tensile properties ($\varphi = 0.5$, $t_H = 60$ s, T = 800/950 °C).

### 3.2.2. Deformation

When using thermal annealing only (cf. micrograph "$\varphi = 0$" in Figure 8a; image analysis results, Figure 8b), precipitation of fine particles was not achievable at 800 °C. Even at short holding times (60 s), all the particles in the "solely annealed" alloy ($\varphi = 0$, in Figure 8) were larger than 800 nm, and most were larger than 1000 nm. The ambient-temperature ultimate tensile strength after "solely annealing" was less than 400 MPa (Figure 8c), which was below even the level of recrystallized material. The coarsened particles thus obtained obviously did not benefit material strength. In fact, the consumption of W and Nb from the matrix additionally decreased solution strengthening and led to the observed drop below the strength level of solution-treated material.

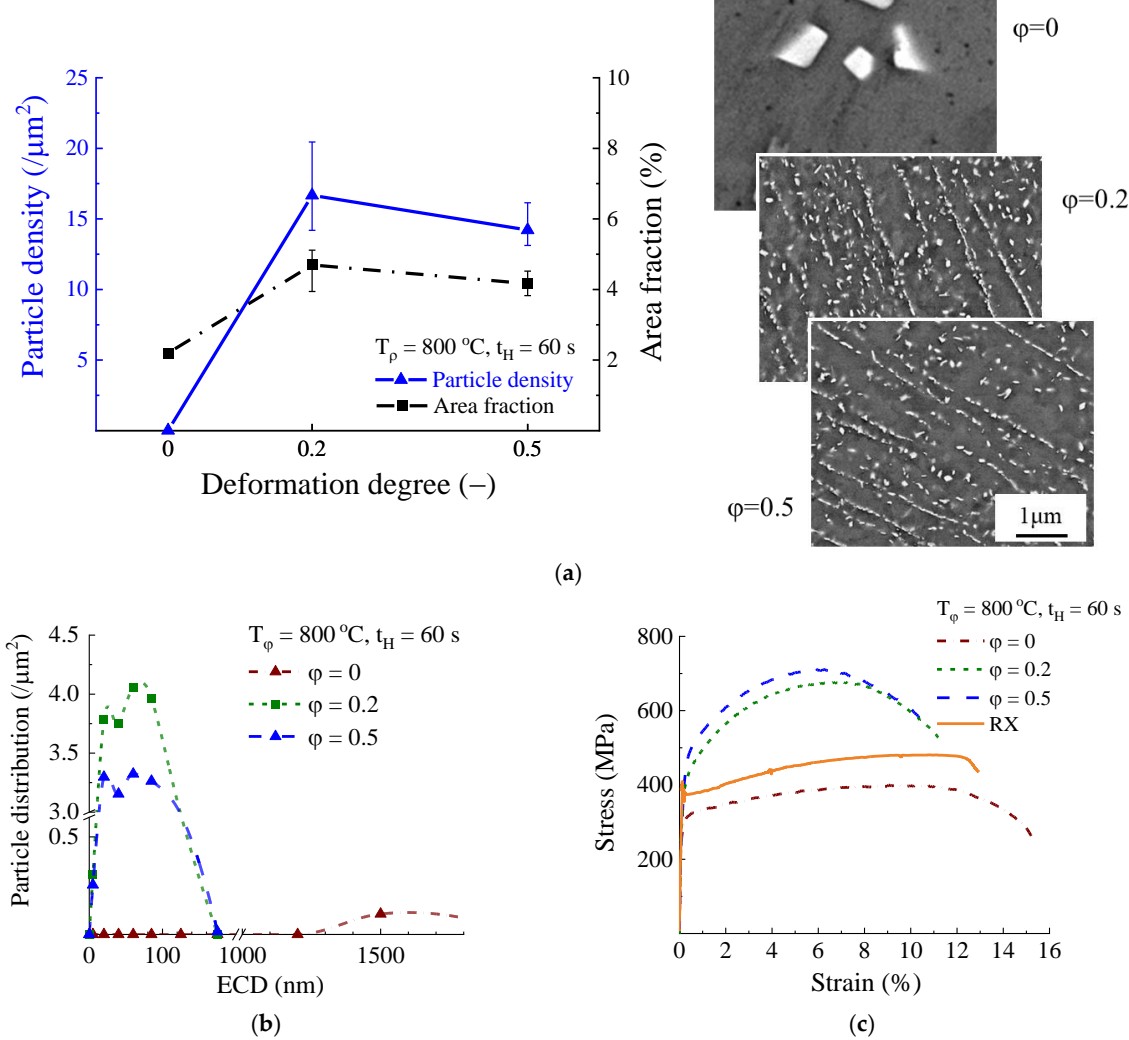

**Figure 8.** Influence of TMP deformation grade on (**a**) particle density and area fraction, (**b**) particle size distribution, and (**c**) ambient-temperature tensile properties ($t_H$ = 60 s, T = 800 °C, $\varphi$ = 0, 0.2, 0.5).

At a processing temperature of 800 °C, a clear trend towards higher tensile strength (Figure 8c) resulted from the introduction of dislocations by TMP (deformation degrees 0.2 and 0.5). Furthermore, it led to increased density and area fraction of Laves-phase precipitates via introduction of additional nucleation sites. The particle size distribution diagrams (Figure 8b) reveal the formation of a multitude of finer precipitates in a size range of generally less than 200 nm, with a dual-peak distribution (around 20 nm and 80 nm), in the TMP material. At higher deformation degrees ($\varphi$ = 0.5), the particle microstructure appeared to be more uniform, with a more pronounced linear arrangement of precipitates than at lower deformation degrees ($\varphi$ = 0.2; cf. Figure 8a). Quantitative analysis did not reflect significant differences concerning particle density and area fraction, which were almost constant within the range of scatter. Consequently, few differences in tensile properties were encountered (Figure 8c). The higher density of dislocations induced by increased deformation might be considered responsible for the more pronounced linear arrangement of precipitates. Nevertheless, the material deformed by $\varphi$ = 0.5 presented slightly higher values for offset yield and ultimate tensile strength (approx. 40 MPa higher than $\varphi$ = 0.2).

### 3.2.3. Holding Time

The influence of prolonged holding time after deformation at 800 °C is depicted in Figure 9. Shifts in particle density, size distribution, and morphology were observed during particle evolution. Immediate cooling after deformation ($t_H = 0$ s) prevented the typical linear arrangement of precipitates (Figure 9a) and resulted in a comparably low density and area fraction of Laves-phase particles, because of the limited time for precipitation. After immediate quenching, the size of the Laves-phase particles presented a wide distribution up to about 400 nm, with a main peak at around 120 nm, but a low particle density (Figure 9b). The highest density of Laves-phase precipitates appeared at 60 s holding time (Figure 9a). Particle area fraction increased only slightly, to about 4%, while the size distribution simultaneously changed dramatically.

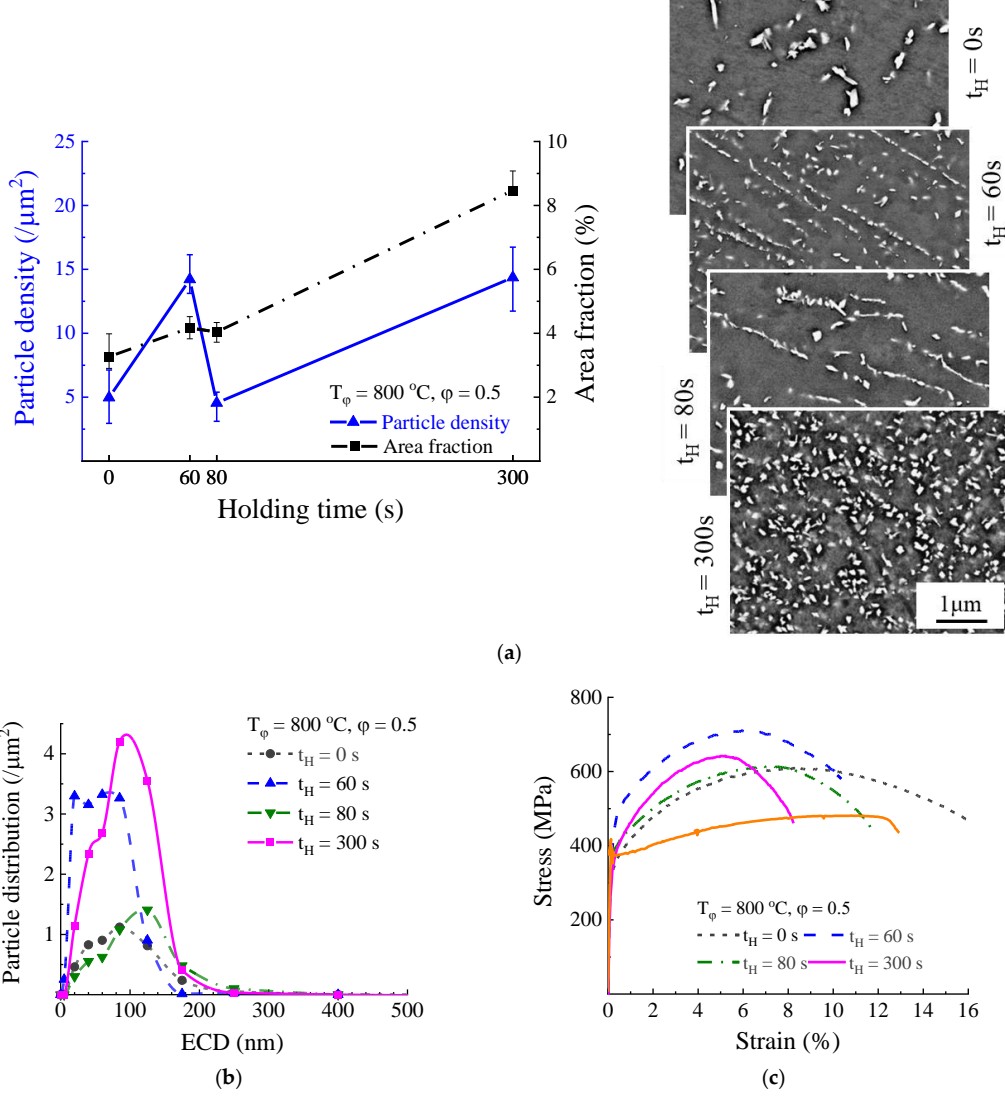

(a)

(b)                                                   (c)

**Figure 9.** Impact of holding time on (**a**) particle density and area fraction, (**b**) particle size distribution, and (**c**) tensile properties (T = 800 °C, $\varphi$ = 0.5, $t_H$ = 0, 60, 80, 300 s).

After a prolonged holding time of 80 s, the particle area fraction remained almost constant at ~4 vol. % (within the range of scatter, cf. Figure 9a). Within 20 s difference (Figure 9b: 60 s to 80 s), coarsening of precipitates changed the size distribution significantly. While at 60 s the size distribution presented a plateau from 20 to 85 nm, the plateau disappeared and a peak was encountered at 125 nm after 80 s of holding. Further prolonging the holding time to 300 s did lead to an increase in

amount density and area fraction (Figure 9a) along with a backshift of the peak in size distribution to about 90 nm (Figure 9b) ECD, and accompanied by a change in particle geometry from preferentially disc-shaped to spherical (cf. Figure 9a, micrographs 80 s vs. 300 s). This might indicate a complex equilibrium of metastable particle species, that were kinetically favored for a short period. Further investigation into this issue is necessary. The differences encountered in microstructure consequently led to differing tensile performance (Figure 9c). The highest tensile strength (710 MPa) was reached with 60 s of holding time, while the other conditions produced tensile strength values ranging around appr. 600 MPa but differed in ductility.

### 3.3. Comprehensive Correlation of Microstructure and Tensile Properties

As discussed above, the mechanical properties of HiperFer steels were strongly influenced by the area fraction, particle density, and size distribution of the strengthening Laves-phase precipitates. Area fraction played a minor role, and particle size distribution and homogenous dispersion the major roles in tensile strength. For this reason, mean equivalent circle diameter values of the precipitates were calculated and compared to the key tensile properties.

The variations in tensile properties can be directly correlated to precipitate microstructures of the model alloys. Figure 10 correlates the individual TMP parameters to the corresponding mean ECD value of the Laves-phase precipitates formed during processing. The general trend towards increased tensile strength with decreasing ECD is obvious in all the diagrams. Decreasing the TMP temperature, increasing the deformation degree, and maintaining the holding time in a reasonable range benefitted size control of the strengthening Laves-phase precipitates and consequently improved the tensile properties that could be obtained.

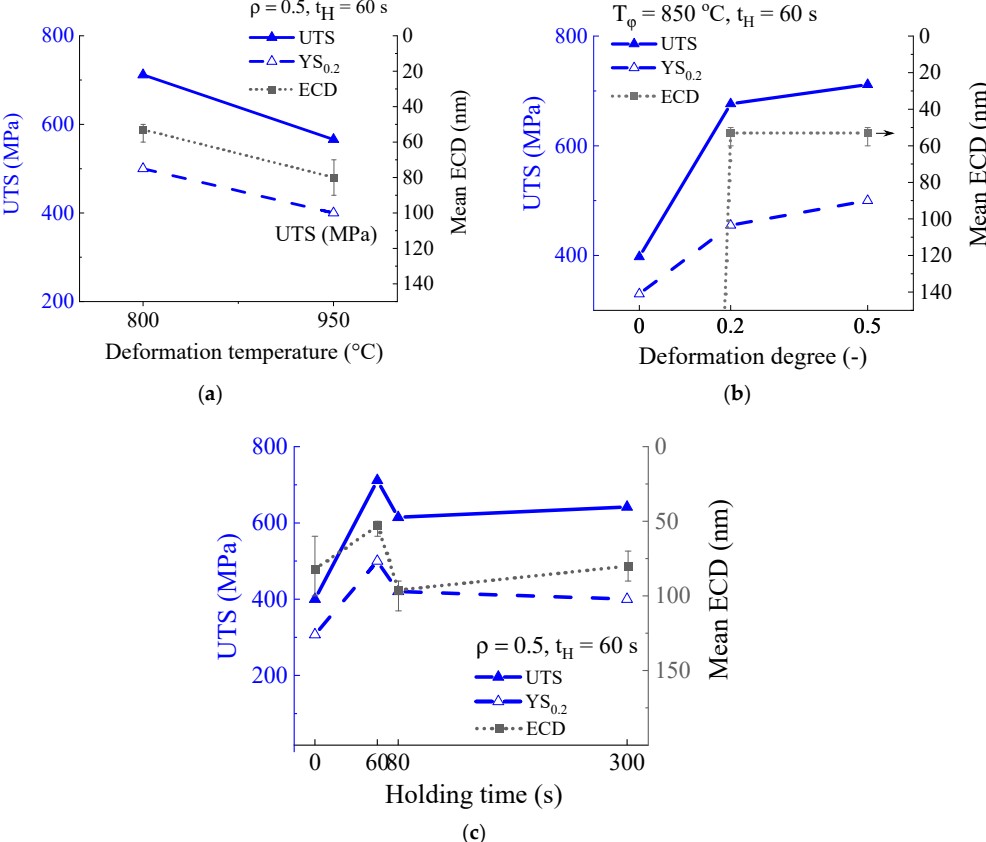

**Figure 10.** Impact of TMP parameters on ambient-temperature ultimate tensile strength and mean ECD of the Laves-phase particles: (**a**) $\varphi = 0.5/t_H = 60$ s, (**b**) $T_\varphi = 800\ °C/t_H = 60$ s, (**c**) $T_\varphi = 800\ °C/\varphi = 0.5$.

On the other hand, the ambient-temperature ductility of TMP HiperFer steel presents a negative correlation to the area fraction of Laves-phase particles. Figure 11 displays the evolution of tensile elongation of the TMP material. The tensile ductility obviously dropped with increasing Laves-phase particle area fraction. This was caused by the high hardness and brittleness of the Laves phase. Increasing the area fraction of a brittle phase decreases the ductility of the ferritic alloy matrix [37,38].

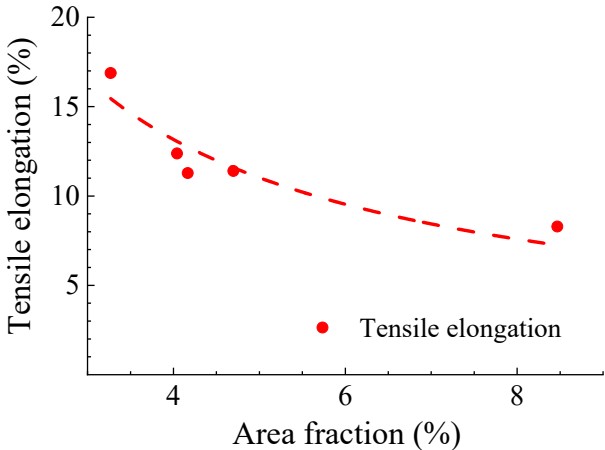

**Figure 11.** Ambient temperature tensile elongation of TMP alloys.

## 4. Conclusions

In the present work, a comprehensive understanding of the interrelations between thermomechanical treatment parameters' effects on microstructure and the resulting mechanical properties of a high-alloyed HiperFer steel were established. The microstructure analysis results confirmed that both area fraction and size of the Laves-phase precipitates can be controlled by varying the thermomechanical treatment temperature, degree of deformation, and holding time after deformation. The mechanical properties were highly dependent on the resulting microstructure and, for this reason, can readily be adjusted during thermomechanical processing. In turn, optimized mechanical properties and more economical production can be achieved via optimized thermomechanical processing.

**Author Contributions:** Conceptualization, B.K., W.B. and X.F.; methodology, X.F., J.P., B.K., M.T. and W.B.; investigation, X.F. and J.P.; resources, B.K. and W.B.; data curation, X.F.; writing—original draft preparation, X.F.; writing—review and editing, B.K., W.B., J.P. and U.K.; visualization, X.F. and B.K.; supervision, B.K.; project administration, B.K., W.B. and U.K.; funding acquisition, B.K. and W.B. All authors have read and agreed to the published version of the manuscript.

**Funding:** Project funding by Deutsche Forschungsgemeinschaft (DFG) under grant number 631895 is greatly appreciated.

**Acknowledgments:** The authors also wish to thank the following staff members of Forschungszentrum Juelich GmbH: E. Wessel, D. Grüner (microstructural examination), B. Werner and H. Reiners (mechanical characterization).

**Conflicts of Interest:** The authors declare no conflict of interest.

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
