# Peer review of "Thermomechanically Induced Precipitation in High-Performance Ferritic (HiperFer) Stainless Steels"

_applsci, doi:10.3390/app10165713_

Round 1
Reviewer 1 Report
The manuscript reports an interesting study on the effects of several thermal treatment parameters on the microstructural and mechanical evolution of a novel ferritic stainless steel.
The topic is interesting and addressing important issues, considering the request of mechanical properties in severe ambient conditions in terms of temperature and atmosphere for a multitude of applications.
Authors studied a novel HiperFer steel and compared a multi-step processing to an integrated thermo-mechanical treatment, promising for the possibility to achieve the same or even better mechanical properties and offering a time- and cost- saving processing approach.
The proposed experimental method is sound and well described, investigating the effect of temperatures, deformation degree and the holding time of the process to the mechanical properties and the microstructural features. Also, the results are clearly presented and thoroughly discussed.
In order to increase the quality of the manuscript, I would like to suggest the following improvements:
- The figure in page 3 (reporting the temperature profiles of the two processing) is marked as Figure 1, however it is the second figure of the manuscript. As a consequence, the following images should be numbered accordingly. Please also check the correspondence of the in-text references to the figures.
- The first two figures are indicated in captions as Figure 1 (in bold), while the following figures as abbreviated Fig. Please refer to the guideline of the journal for the proper caption formatting.
- In caption of Fig.3 (page 6), Authors should consider to specify the type of used microscope and the detector mode (SEM-BSE?)
- The caption of the figure at the bottom of page 5 is missing (between fig. 2 and fig. 3)
- In page 6, lines 155-159, Authors describe the transition between a IGexterior and IGcenter microstructure, characterized by different size and distribution. Have the Authors found a homogenous thickness of the IGexterior area? Is this thickness somehow influenced by the tested parameters? Maybe it is worth taking in consideration also this feature for discussing the effects on the mechanical properties.
- In Figure 7b and c; 8c; 9a-b-c not all the curves are clearly visible. It is therefore suggested to improve the quality of the figures or to increase the thickness of the lines.
Author Response
Thank you very much for your valuable comments.
For our changes refer to the following list of answers to your question and the attached, marked up manuscript.
List of comments/answers:
- The figure in page 3 (reporting the temperature profiles of the two processing) is marked as Figure 1, however it is the second figure of the manuscript. As a consequence, the following images should be numbered accordingly. Please also check the correspondence of the in-text references to the figures.
Re: All the figures and in-text references to the figures were corrected
- The first two figures are indicated in captions as Figure 1 (in bold), while the following figures as abbreviated Fig. Please refer to the guideline of the journal for the proper caption formatting.
Re: Formatting was corrected.
- In caption of Fig.3 (page 6), Authors should consider to specify the type of used microscope and the detector mode (SEM-BSE?)
Re: added
- The caption of the figure at the bottom of page 5 is missing (between fig. 2 and fig. 3)
Re: This is not an individual figure, but a schematic drawing to illustrate the locations of IGc and IGe areas, mentioned in the following. Some changes have been implemented into the figure: “Microstructure schematic” has been added at the left of the top line and the schematic is now mentioned as the top line in the figure caption. The formerly “top” line has been renamed to “middle”. We hope that this issue is now cleared.
- In page 6, lines 155-159, Authors describe the transition between a IGexterior and IGcenter microstructure, characterized by different size and distribution. Have the Authors found a homogenous thickness of the IGexterior area? Is this thickness somehow influenced by the tested parameters? Maybe it is worth taking in consideration also this feature for discussing the effects on the mechanical properties.
Re: This is a very good point.. congrats! In fact, there are variations. Unfortunately there is no clear interrelation to the variations in processing parameters. We suspect that grain orientation and / or possibly other (still unidentified) parameters may play a role. This is a point of concern and will be tackled in future research.
- In Figure 7b and c; 8c; 9a-b-c not all the curves are clearly visible. It is therefofore suggested to improve the quality of the figures or to increase the thickness of the lines.
Re: Figures have been revisited. We tried to do the best concerning this point.
Thank you very much for your valuable review. To make the changes easier to follow we added a PDF with all the changes marked-up.

Reviewer 2 Report
- Please re-order all the figures, there are two Figure 1 in the manuscript.
- In the second Figure 1, what does "Td" mean? Please specify. If it was deformation temperature, what is TΦ in Line 92, please unify them all over the manuscript.
- In the caption of Figure 3, the "particle free zones (PFZS)" is not so accurate, should be "precipitate free zones (PEZS)".
- In my opinion, "exterior areas" means areas outside the grain, but according to Figure 3, the authors did not mean the areas outside the grain, should be the area near the PFZ area, so please correct the corresponding descriptions all over the manuscript.
- According to the title, this study should focus on the difference between MSP and TMP, so I think it is not so necessary to compare MSP 1 & TMP and MSP 2 & TMP. Just try to clarify clearly one difference should be enough, the current version seems a little messy.
- Please unify the legend in Figure 6. There is "Φ=0.5" in the sub-figures, but "ρ=0.5" in the caption. One more thing, Figure 6c has been cut too much, pay attention to the x-coordinate.
- Line 135, the author claimed the grain size is ~800 to 1500 μm, and Line 116, the sample is 10*2*1 mm3 in the gauge dimension, therefore, the sample is about 1-grain thickness. According to Figure 5, the precipitation of the Laves phase is orientation-dependent, I was wondering when the sample is not thick enough, whether there is any deformed grain structure of tested sample, whether there is any preferential orientation.
- I would like to suggest going through the manuscript more carefully for clarity, syntax and correctness. The English should be greatly improved for the sake of clarity.
Author Response
Thank you very much for your valuable comments.
For our changes refer to the following list of answers to your question and the attached, marked up manuscript.
List of comments/answers:
- Please re-order all the figures, there are two Figure 1 in the manuscript.
Re: Done
- In the second Figure 1, what does "Td" mean? Please specify. If it was deformation temperature, what is TΦ in Line 92, please unify them all over the manuscript.
Re: “Td: Dissolution temperature of the Laves phase”. A statement has been added to the figure caption
- In the caption of Figure 3, the "particle free zones (PFZS)" is not so accurate, should be "precipitate free zones (PEZS)".
Re: Particle (or precipitate) free zones at (high angle) grain boundaries are a specific feature of this type of alloy and the acronym “PFZ” was used in a row of papers throughout the last years. For consistency reasons we´d like to maintain “PFZ” rather than switching to “PEZ”. Hope this is o.k. for you.
- In my opinion, "exterior areas" means areas outside the grain, but according to Figure 3, the authors did not mean the areas outside the grain, should be the area near the PFZ area, so please correct the corresponding descriptions all over the manuscript.
Re: An area “outside a grain” would be inside another grain, or the surrounding atmosphere. Our expression might not be the most elegant, but it does the job. To clarify what we meant we implemented the microstructure schematic in the top line of Fig. 4 (corrected numbering).
With this it is obvious that “exterior” areas belong to the intragranular area, but are located close to the PFZs and thus are not “outside” the grain, i.e. inside a neighbor grain or the surrounding atmosphere.
- According to the title, this study should focus on the difference between MSP and TMP, so I think it is not so necessary to compare MSP 1 & TMP and MSP 2 & TMP. Just try to clarify clearly one difference should be enough, the current version seems a little messy.
Re: This may more be a question of personal taste. The reasons why we implemented two variants of standard multistep heat treatment and thermomechanical processing each is to show the bandwidth of obtainable properties. And that both treatment options can yield properties which are not too different.
But we agree: It makes the paper more demanding to read and understand.
- Please unify the legend in Figure 6. There is "Φ=0.5" in the sub-figures, but "ρ=0.5" in the caption. One more thing, Figure 6c has been cut too much, pay attention to the x-coordinate.
Re: Both done
- Line 135, the author claimed the grain size is ~800 to 1500 μm, and Line 116, the sample is 10*2*1 mm3 in the gauge dimension, therefore, the sample is about 1-grain thickness. According to Figure 5, the precipitation of the Laves phase is orientation-dependent, I was wondering when the sample is not thick enough, whether there is any deformed grain structure of tested sample, whether there is any preferential orientation.
Re: In the presented specimens precipitation took place during either multistep heat treatment (MSP) or thermomechanical treatment (TMP), not in miniature specimen testing.
TMP was carried out at “Specimens of 15 x 15 x 65 mm3 in size for simulated TMP” … , which were … “cut from the homogenized blocks.” This was stated in section 2.1. The miniature specimens for tensile testing were then machined from the deformed sections of the specimens, applied for simulated (i.e. small scale, to consume less material) TMP. This ensured deformed grain structure.
- I would like to suggest going through the manuscript more carefully for clarity, syntax and correctness. The English should be greatly improved for the sake of clarity.
Re: The manuscript was re-checked for figure numbering and in-text references to figures. Thank you very much for pointing out this issue. The topic and all the specific details make the paper demanding to read and understand. With this in mind we invested a lot of time to be as precise and clear in descriptions as possible and on the other hand to avoid improper simplification…
For this reason it feels more than a little embarrassing, that the figure numbering issue occurred. Sorry for this.
Language and style are the result of the cooperation of three experienced English speakers. With the complexity in mind we think we have been as clear as possible.
Thank you very much for your valuable review. To make the changes easier to follow we added a PDF with all the changes marked-up.

Round 2
Reviewer 2 Report
Accept